# Development of Sustainable Test Sites for Mineral Exploration and Knowledge Spillover for Industry

**Michaela Kesselring** [1,*] **, Frank Wagner** [2] **, Moritz Kirsch** [3] **, Leila Ajjabou** [3] **and Richard Gloaguen** [3]

[1] Institut for Human Factors and Technology Management, University of Stuttgart, Nobelstraße 12, 70569 Stuttgart, Germany

[2] Fraunhofer Institut for Industrial Engineering; Nobelstraße 12, 70569 Stuttgart, Germany; frank.wagner@iao.fraunhofer.de

[3] Helmholtz Zentrum Dresden Rossendorf, Helmholtz Institute Freiberg for Ressource Technology, 09599 Freiberg, Germany; m.kirsch@hzdr.de (M.K.); l.ajjabou@hzdr.de (L.A.); r.gloaguen@hzdr.de (R.G.)

**\*** Correspondence: michaela.kesselring@iat.uni-stuttgart.de; Tel.: +49-711-9702172

**Abstract:** In mineral exploration, pressure is growing to develop innovative technologies and methods with a lower impact on the social and physical environment. To assess the performance and impact of these technologies and methods, test sites are required. Embedded in the literature on sustainable development, this paper explores how social and environmental measures can be implemented in the design of test sites and what industry stake can learn from sustainable test sites. Through qualitative research, two value networks were developed, one for a sustainable test site approach and another for the existing business practice in mineral exploration. Respondents include public sector officials as well as experts in the social, environmental, business, geoscience, and industry fields. The analysis identifies key drivers for the development of socially and environmentally accepted test sites, thus drawing up actionable points for the mineral exploration industry to increase sustainability. The findings of this paper suggest that the integration of experts and partners from social, as well as environmental, sciences drives sustainability at test sites. For industry application, this results in the need to adapt the activities performed, align resource use with sustainability indicators, and also reconfigure the network of partners towards more socially and environmentally oriented business practices.

**Keywords:** mineral exploration; sustainable development; test sites

## 1. Introduction

Sustainable economy has risen to the top of political and economic discourses, capturing media attention and sparking controversy in an era of heightened awareness for anthropogenic impacts and scarcity of the Earth's natural resources [1,2]. Perhaps unsurprisingly, traditional powerhouse industries such as the automobile sector and natural resource extraction have been scrutinized regarding their contribution to the protection of the environment, or rather the lack thereof [3,4]. Mining is under pressure in this context due to the large environmental footprint in the extraction process, as well as the invasive practices in the exploration of minerals [5,6]. Consequently, there are nascent efforts to develop less invasive techniques, defined as a set of energy-efficient, low-impact, and socially acceptable techniques that improve mineral targeting [7,8]. Arguably, the costs of changing a component of a technology or technique increase over the course of the technological development process. Therefore, sustainability considerations are preferably made at an early stage in the research and development process [7,9]. Besides existing eco-tools for sustainable product development, as presented by [10],

there is little literature on how to promote sustainable technologies while also pushing sustainability with respect to social and environmental acceptability. We contribute to this strand of literature by exploring the case of test sites in mineral exploration. In theory, technology tests in natural environments provide a platform for assessing the interplay between technology, social, and physical environment while boosting technological development. Testing in natural environments is a complex and expensive process, which often exceeds the capabilities of a single company or even industry [11]. However, test sites or reference sites are by nature experimentation platforms that allow simulating different parameters on a small scale, with the goal of translating lessons learned into practical applications. Hence, rationalizing sustainable development at reference sites can reverberate in sustainability lessons of related sectors. This reinforces the importance of reference sites and strengthens the need to investigate the relationship between reference sites and current industry practices.

In recent years, public authorities have recognized the value of reference sites and subsequently increased funding on related topics. One prominent state actor is the European Union (EU), where these sites are referred to as "pilot locations" in calls for proposals. With several publicly funded projects underway, there are plenty of theoretical concepts that address the harmonization of economic, environmental, and social requirements [12]. At present, there is room for including the technological perspective into research and provide practical insights on the interplay of technology and the social and physical environment. Still, more research is required on how results from reference sites can be transferred beyond the technical perspective and into broader sustainability frameworks. Hence, two aspects are under-represented in the current debate around sustainable development: (1) a system-oriented approach that investigates how sustainability assessment can be included in the technological development process as early as the testing phase, and (2) how industries can learn from sustainable reference sites.

The purpose of this paper is first to examine the interaction between technology, the social and physical environment to understand how to design test sites that meet sustainability needs. Per definition, sustainability" meets the needs of the present without compromising the ability of future generations to meet their own needs" [13] (p. 43), which reveals the need to study linkages between the blocks that form the environment. Accordingly, the term sustainable reference site is employed to refer to a test site that integrates social and environmental aspects in the technology development process. In a second instance, the paper explores how the insights gleaned from sustainable test sites can promote sustainability efforts in the mineral exploration industry by carrying out a case study.

Therefore, the article responds to the following research questions (RQs):

1. How can social and environmental measures be implemented in the design of reference sites?
2. What insights can sustainable reference sites provide for companies aiming to improve their sustainability efforts?

The RQs are examined with a qualitative research approach as explained in the methodology section. For a practical reference point, the EU-funded project INFACT (Innovative, Non-Invasive and Fully Acceptable Exploration Technologies; this project has received funding from the European Union's Horizon 2020 research and innovation program under grant agreement No 776487), serves as a use case. However, the research approach is explicitly developed to be transferable beyond this case. INFACT aims to establish a reference site infrastructure for the development of innovative non-invasive, socially accepted exploration technologies, and thereby, promote the fair chain of raw materials [14]. In doing so, INFACT follows an inclusive approach, connecting researchers from social, environmental, and geosciences with technology developers, as well as mineral exploration companies and political stakeholders. The project was chosen as it specifically targets the integration of environmental and social aspects into the technology development phase. Based on the insights gained, a way for integrating sustainability efforts into technological development was identified for INFACT. Furthermore, improvement opportunities for a more sustainable mineral exploration industry were identified and analyzed. The results show that the main drivers of sustainability for technology

development are related to the integration of experts from social and environmental sciences, and that the same holds for industry practices beyond technology development.

In summary, the study develops a sustainable test site approach and aims to translate the outcome into actions for higher sustainability in the mineral exploration sector. The implementation of leverage for sustainability in the technology development process and translation of the insights into business practice reinforces the importance of the present study. This paper addresses current challenges, given that recent data reveals that the mineral industry struggles with a negative image mainly related to sustainability issues [15].

## 2. Sustainability in Mineral Exploration

Several definitions of sustainability exist. The most common defines sustainability as answering to the requirements of today without jeopardizing future requirements [16]. The authors [17] broaden this perspective and define sustainability as the "potential for long-term well-being of the natural environment, including all biological entities as well as the interaction among nature, individuals, organizations and business strategies" [17] (p. 400).

To realize this definition of sustainability, the UN proposed 17 sustainable development goals (SDGs) in 2015. Among those is the goal to increase industry, innovation, and infrastructure [18]. The successful realization of this goal requires the continuous supply of raw materials [19]. While recycling and innovation can reduce the total use of raw materials, the dependency on raw materials will remain of concern for the near future [19,20]. In consequence, responsible sourcing of minerals stands at the very beginning of sustainable development. Concrete actions for responsible sourcing, and securing the SDGs in the mineral resource sector are the creation of cross-sectoral [21] and cross-country cooperation [22], strong home state regulations that ensure sustainable operations [23] as well as shared value management system [24,25]. The adoption of these actions varies between companies [26,27], countries [22], and industry associations [28]. On both the company and industry association level, the minerals sector often stands in contrast to economic and political self-interest [26,28–30].

The research to date has tended to focus on the resource extraction and processing phase of the mineral-based product life cycle [31]. Availability of minerals today and in the future, however, starts in the exploration phase. Here mineral systems research, as well as exploration methods development, form the foundation of all value creation in the minerals sector [32]. Potential interventions in mineral exploration go beyond mitigating the impact of mining but towards building new links between social engagement and environmental practices for inclusive development solutions [19,33]. The present paper adheres to these approaches and links inclusive technological development solution at mineral exploration technologies test sites with approaching sustainable development across the entire industry, for long-term well-being.

### 2.1. Performance Testing in Real Environments

Especially for technologies where the end setting is a natural environment, field tests are crucial to verify theoretically modeled performance estimations and derive improvement potential [34]. According to [11] field tests are "tests of technical and other aspects of a new technology . . . in a limited, but real-life environment" [11] (p. 3). Associated benefits are product performance improvements [11] or accelerating commercialization by demonstrating the capabilities of a technology [35].

In mineral exploration, testing geophysical and remote sensing technologies in real-life test scenarios can be extremely complex. As the performance parameters vary with diverse physical or technological properties, different agendas for testing need to be identified [36–40]. In general, test sites in the field of mineral exploration are used for proof of concept, robustness and accuracy tests, calibration and validation. Exemplary for this are the established test sites in Denmark [38] and Italy [39] that serve to study time-domain electro-magnetics (TEM). These sites target the calibration and validation of systems and enable the verification of the processing and inversion of measured data [38–40], but also help with robustness and accuracy testing [39]. Other test sites, such as the

Canadian Reid Mahaffy, specialize in technologies that allow for geophysical airborne methods [36,37]. Furthermore, sub-surface imaging can be improved by comparing technologies with different maturity levels [37]. According to [41] performance improvement can further be realized through repetitive benchmarking of a new system against the previous system's configuration in the same test area.

This indicates that test sites are either employed for performance improvements or for demonstrating the performance of technologies to third parties. Particularly in cases where the performance is evaluated by third parties, the reliability of the technology increases, and so does the visibility and ability to reach potential customers [42].

While test sites generally promote technological development and commercialization, there is a gap in the literature for a more systematic approach to incorporate sustainability into test sites. In the present political climate that demands increased sustainability, two measures are crucial. On the one hand, there is a need to minimize the invasiveness of exploration through innovation in order to reduce the footprint in environmentally sensitive areas. On the other hand, going beyond the technological perspective, [43] claim that field trials that solely focus on technological development are less successful than those embedded in a socio-technical system. This relates to a more general notion of [44] (p. 1), which suggests that "effective stakeholder engagement can improve the environmental and social sustainability of projects". Hence, the assessment, management, and monitoring of the social and physical environment are crucial to improve sustainability measures.

By analyzing the existing literature, it becomes clear that sustainability is merely considered in the context of efficiency gains of a technology or the structure of the pre-built environment [35,45,46]. However, there is an opportunity to utilize field trials to assess the performance of technologies while also analyzing the potential impact on the social and physical environment [11]. At present, this remains a theoretical construct. According to the literature analyzed, in mineral exploration, no such test site has yet opened for commercial use.

To bridge the gap and stimulate sustainability in technology management, this paper builds on the notion that test sites represent a network of organizational actors, who operate within social and physical environments. Authors such as [47] support this notion and argue that technological development can be coordinated, by strategically designing the network in which a technology is embedded in. Translating these insights into requirements for test sites, sustainability in technology development may be fostered by aligning the value network of a test environment with social and environmental standards. One way of illustrating the value network is through business models. Here, the business model serves as an analytical framework that helps coordinate the value network of test sites. Value is then defined as the relative importance of a business model element (e.g., service, product, activity, resource) [48]. The following part of the paper describes in detail how existing research on business models views the topic of sustainability and can add to designing sustainable test sites.

## 2.2. Sustainable Business Networks

Business models that deal with different aspects of value-adding mechanisms are crucial for the conceptualization and analysis of test sites. One of the most commonly cited definitions for value-adding systems is the work of [49] (p. 14), which describes a business model as "the rational of how an organization creates value". A more technologically oriented consideration comes from [50], who investigated the success factors of technology spin-offs.

Following the technological train of thought, [51] claim that by aligning business models with sustainability measures, systemic changes in the technological landscape can be realized. By demonstrating that the full potential of technologies can only be realized when the respective technology is embedded in a value-adding network, [52] further spur this discussion.

Business models are originally oriented towards promoting a technology, product, or service. In recent years, a number of studies have begun to examine how sustainability efforts can be integrated into this approach [53]. At the center of consideration is the notion that value creation is no longer

only driven by a product, but takes place in a collaborative way with peers from society and environment [54,55]. Firms acting together and joining forces with external parties through informal or formal arrangements [54] require more holistic approaches towards value creation than the narrow technological focus presented earlier [54,56]. According to [46] (p. 4) a sustainable business model as a concept to connect a company's activities "to the larger systems of which they are part." In the literature on sustainable business models, the focus varies by industry [57,58], country [59,60] or more general the scope considered [53,61,62].

To this point, existing accounts fail to resolve the frictions between sustainability and technology development at test sites. Turning to the RQ, the two key challenges are to design a value system that not only promotes technical considerations, while also capturing the stakeholder interests with regard to society, environment, or economics. On the other hand, and in order to derive lessons learned for industry partners, the system must allow for compatibility with the industry at large.

For value system design in general, there are various frameworks that describe the compositional factors that constitute a value network. Depending on the framework, authors refer to these elements as functions [50], components [63], or building blocks [49]. Irrespective of the terminology used, business model frameworks share the objective of defining these factors and their relationships in order to describe "how organizations can create and capture customer value" [50] (p. 99). The complexity, and thus, the ease of use of existing frameworks strongly differs. While the triple-layered sustainable business model canvas introduced by [51] comprises 27 blocks, the business model canvas of [49] with nine building blocks is less complex and therefore more applicable for audiences with different backgrounds and varying levels of business knowledge.

The business model canvas of [49] is displayed in Figure 1. Here, the left side of the business model canvas describes how value is created. The value creation consists of "Key Resources" such as know-how, facilities or manpower required for a company to create value, "Key Activities" needed to generate value, and "Key Partners" that form stakeholder networks. The value proposition is placed at the center, as it describes the offer made to the customer. Finally, the right side of the canvas describes how value is captured. It contains "Customer Segments", defining the customer groups that are relevant to the business. "Channels" are employed to address the above "Customer Segments" as well as "Customer Relationship" which describe the types of relationships the business has with its customer segments. Fourth, the business model canvas consists of the building blocks "Revenue Streams" and "Cost Structure". "Revenue Streams" describe the way a company generates revenues and the "Cost Structure" characterizes the costs involved in creating value [49]. The canvas, as proposed by the authors, is displayed in Figure 1.

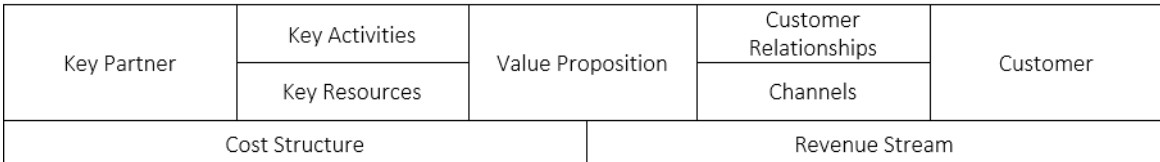

**Figure 1.** Business model canvas (own representation aligned [49]).

Regardless of the specific framework used to structure the building blocks, a key concern for the sustainability challenge is the connection between the blocks or elements in a globally networked environment [64,65]. The business model canvas identifies the main elements to be addressed for sustainability, but it is not sufficient in defining the path to reap value from them because it considers each block a separate entity. Disregarding interdependencies between structural constructs simplifies the complexity of systems and aggravates risk assessment and the planning [66,67]. Therefore, a second layer is needed to explicitly address the linkages between the blocks. One system that lends itself to this task is asset management, which has been proposed specifically in the mining sustainability context already [65]. Asset management refers to "a coordinated activity of an organization to realize value from assets" [68]. According to [65], coordinated value system management can increase the

sustainability return of each building block, through planning and prioritization of assets [67] and reduce the operational risk [29,30,68]. The existing literature points to several reasons why asset management, in particular, is useful for mining. Both [29,30] discuss applications for risk management, reference [65] identifies asset management as a way to manage complexity across countries, and mining sites, and [69] identified increasing the financial sustainability of a business as a result of strategic asset management. Both, [69] as well as [70] further found environmental and social sustainability improvements through asset management. As a result, [65] argues that robust, sustainable, and resilient business across different sites can be built.

Thus far, this paper presented a critical assessment of the recent literature on sustainability, mineral exploration test sites, and business models. It was recognized that concrete plans for sustainability-oriented operations and technical development exists that have the potential to transform systems towards sustainability. In what follows, the paper introduces how data in this paper were collected to analyze the relationship between technical development at test sites, business models, and sustainability in mineral exploration.

## 3. Case Description

The central data in this paper were collected through in-depth investigation and examinations of the EU-funded project for innovative non-invasive and fully acceptable exploration technologies (INFACT). The project was chosen as the central source for sampling as it links technological development opportunities to country-specific social and environmental contexts and allows for the creation of benefit-sharing value creation and implementation.

INFACT has undertaken a far-reaching process of increasing the availability of real-life test sites, with the main value proposition of unbiased third-party approval. The test site infrastructure built up by INFACT consists of four reference sites. The sites are located in Rio Tinto and Cobre las Cruces in Spain, Sakatti in Finland as well as Geyer in Germany [14]. The Spanish sites have volcanic-hosted massive sulfide deposits, mined mostly for copper, but also zinc, lead, and to a lesser extent, gold and silver [71,72]. They are located in a Mediterranean climate [73]. The Finnish site has a copper-nickel-platinum group elements deposit and is located in the subarctic climate [74]. The German site has tin bearing deposits also containing tungsten, zinc, molybdenum, copper, iron, silver, and indium [75,76] and is located in a humid continental climate [77].

In addition, a range of social relationships is represented by the three test regions. In southern Spain, mining operations are supported for millennial and the communities live in close proximity to active mines [14,78]. The region around Geyer has seen a long history of mining and is located close to a wildlife park [14,79]. Concerning the sparsely populated test site region of northern Finland, sustains mining as well as tourism and reindeer herding [14,80,81]. The finish test site is on the edge of a protected "Nature 2000", and the Sámi homeland [14,81].

Foremost, the project aims to provide interested parties with different geological settings and styles of mineralization. In addition to the diverse geology and climate, the reference sites differ in vegetation, wildlife, and population density, which allows for the evaluation of varying social and environmental factors. In addition, the sites are situated in very different socio-economical environments and have varying degrees of experience with exploration and mining [14].

INFACT recognizes that the success of the reference sites lies in promoting the interplay between the technical, social, and environmental aspects determining the value network of the reference sites [14]. Reviewing the requirements towards a more sustainable mineral exploration practice, the project satisfies all apparent requirements for sustainable technology development.

Increasing sustainability in mineral exploration means that the invasiveness of technologies, methods, and systems should be minimized through innovation. In addition, the effective integration of social and environmental stakeholders can further boost the sustainability of projects [44] (p. 1). Hence, the selection of INFACT as a case to develop and analyze sustainable test sites is deemed appropriate for answering the RQs.

## 4. Methodology

The objective of this study is to gain a deeper understanding of the changes induced by adopting a sustainable development approach at natural test site environments and to translate insights into business practice. The study is characterized as exploratory since there is no sufficiently integrative and systematic approach to develop and analyze sustainable test sites and draw conclusions for industry practice. For exploratory studies, [82] suggests a qualitative research approach because a quantitative approach hardly conveys in-depth information. For studies involving the opinion of different stakeholders, [83] further propose workshops as a qualitative data sampling method. According to the authors, workshops enable the collaboration and interaction between different parties and have a positive impact on the credibility of interdisciplinary studies [84]. Hence, a qualitative approach was chosen, whereby workshops served for data generation.

In sum, three workshops were conducted. Participants in the first two workshops were selected from the INFACT expert consortium. The sample comprised two representatives from social sciences, one from environmental sciences, three from geosciences, and two with an engineering background, one industry representative, two government officials, and one economist. Prior to the interview process, the participants had already worked on the INFACT project for 10 months, thus giving them substantial knowledge on the socio-economic context and the particularities of testing in real-life environments. All participants were experts in their field, where an expert is characterized by definite and sound knowledge about a certain area or topic as well as by exclusive access to information [85]. Apart from their expertise in the field, the panel of experts is international such that all countries in which the test sites are located are represented by natives. Three experts are from other countries, so the sample is not exclusive to domestic stakeholders.

The third workshop had the objective of integrating participants from industry, communities, and environmental sciences into the ideation process that are not involved in the INFACT project themselves. Here, nine participants with economic interests and nine participants with an interest in the conservation of the social and physical environment took part. Criteria for selecting the participants included their ability to provide rich information about the topic, hence their interest as well as the affectedness of sustainability in the mineral exploration sector and the development of test sites. This approach was taken to reduce selection bias, secure the validity and public relevance of the study while facilitating more adequate statements about the RQ's. While the ultimate goal of the workshops was the development of a general test site approach as a conglomerate of three test sites, the regional idiosyncrasies were discussed during each workshop, and location-specific adaptations were incorporated.

Given the diverse backgrounds of the participants and the time constraints of the workshops, the business model tool had to be simple yet comprehensive. Therefore, the business model canvas by [33] was identified as the most suitable tool. The canvas chosen for two reasons. First, its popularity in the industry and widespread adoption. Second, it is simple yet comprehensive and thus allows for cooperation across disciplines with varying knowledge levels [65]. The workshops were consequently structured according to the nine building blocks of [49]. To ensure comparability and secure the validity of results, all participants were initially introduced to the logic of the canvas [50]. The workshops lasted approximately three hours. The first workshop was geared towards investigating the business model of current exploration operations. The second and third workshops then built on the first one and had the purpose of investigating the factors that determine the value network of a sustainable test site.

After all, workshops were completed, the data collected on blueprints of the business model canvas was transferred into written documents, and statements with similar notions were synthesized. For answering the RQ 1, the aspects identified in all nine building blocks of the sustainable test site approach were analyzed. To answer RQ 2, the test site business model and the mineral exploration business model were compared.

## 5. Results

Table 1 presents the results obtained from the blueprints of the canvases. In the table, the factors identified for the sustainable test site are listed on the right, while the left side of the table illustrates the existing value network in mineral exploration. For reasons of simplicity, factors that were named in both value networks are listed in the same row.

**Table 1.** Test Site and Mineral Exploration Network, Study Results.

| Factor | Test Site Value Network | Mineral Exploration Value Network |
|---|---|---|
| **Value Proposition** | Social Economic Development<br>Operate in areas with a high degree of social acceptance/Obtain Social license to operate in the areas<br>Ability to obtain expert feedback on social challenges in sensitive areas and coping mechanism thereof<br>Obtain performance data for the improvement of a technology alongside the technological readiness level (TRL)<br>Obtain access to a well explored area that meet current challenges (e.g., tailings)<br>Ability to produce case studies<br>Reduce drilling<br>Support chain of fair EU raw materials<br>Promotion of non-invasive technologies<br>Work in a safe and monitored environment<br>Ability to obtain expert feedback on environmental challenges in sensitive areas<br>Improve discovery rates<br>Ability to publish data gained (Demonstration Platform) | Improve discovery rates<br><br>Increase share price<br>Insights about the subsurface structure (models)<br>Early engagement in promising exploration activities<br>Get access to technology |
| **Key Activities** | Public relations/communication<br>Engagement (communities, sustainability)<br>Monitoring sites (social, environmental activities)<br>Benchmark for new technologies/techniques<br>Evaluation and assessment of technological performance with respect to social and environmental aspects<br>Evaluation of exploration technologies and systems<br>Educational programs for communities | Public relations/communication<br><br>Geoscientific activities (especially drilling)<br>Expectation management towards shareholders<br>Data management |
| **Key Resources** | Social license to operate<br>Contract intangible assets<br>Accurate and precise knowledge and data about the subsurface and the social and physical environment<br>Experts (geosciences, environmentalists, conservationists, social scientists) | Contract intangible assets<br>Accurate and precise knowledge and data about the subsurface<br>Experts (geoscientists, engineers, land-use, economist)<br>Technology |

**Table 1.** *Cont.*

| Factor | Test Site Value Network | Mineral Exploration Value Network |
|---|---|---|
| **Key Partners** | Social and environmental advocates<br>Independent agencies<br>Governments/ politicians (national, regional, local level)<br>Expert association (social, environmental, technical experts)<br>Mining companies<br>Non-profit organizations<br>Academia<br>Locals (e.g., land-use experts)<br>Legal peers (contractor, regulators) | Governments/ politicians (national, regional, local level)<br>Expert association (technical, economists)<br><br><br><br>Locals (e.g., land-use experts)<br>Legal peers (contractor, regulators) |
| **Cost Structure** | Cost of work<br>Database and storage<br>Expert consultation<br>Contract fees (land-access, permits, royalties)<br>Communication with peers and from social and environmental backgrounds as well as social engagement | Cost of work (especially drilling)<br><br>Expert consultation<br>Contract fees (land-access, permits, royalties) |
| **Customer Relationship** | Trust and transparency<br>Unbiased expertise and protocols | Trust and transparency<br><br>Constructive journalism |
| **Customer Segments** | Technology provider and developer<br>(Junior) Mineral exploration companies | Mining companies<br>Shareholder |
| **Distribution Channels** | Media (online/offline)<br>Conferences<br>Case studies as a way to communicate technological performance<br>Direct communication (talks, calls) | Media (online/offline)<br>Conferences<br>Case studies as a way to communicate technological performance<br><br>Stock exchange |
| | Site access fees<br>Valuation of performance (technical, social, environmental) | External investment<br>Share price<br>Project sales/mineral sales |

In order to analyze and draw conclusions from the sustainable test site approach and the value network of the traditional mineral exploration industry, the key elements of the value networks are presented and analyzed in the following.

**Value Proposition:** Based on the workshop output, the existing value proposition of mineral exploration was identified. Here, it becomes clear that the focus of the added value is based on the economic perspective. This includes non-monetary values such as increased knowledge about the subsurface, improvement of discovery rates, and early engagement in promising projects. Monetary aspects, such as increasing the share price were also identified. Regarding the sustainable test site approach, the central value of test sites is the possibility to access and conduct measurements in an area with a well-established subsurface structure. The test site further offers third party evaluation of the conducted measurements to determine the accuracy or precision and fitness-for-purpose. As a result, technology providers can identify performance improvement potential. From an economic perspective, the third-party evaluation can be used to demonstrate the capabilities of a technology to customers or investors. By comparing the measurements with existing benchmarks, the level of impact technologies have on the social and physical environment can be analyzed. Additionally, the test sites aim for being accepted by stakeholders, communities, employees, and the public. To increase social and environmental compatibility beyond the test sites, social and environmental acceptance consulting is offered.

**Key Activities:** Out of the sequence of activities associated with mineral exploration activities, the participants identified geoscientific activities as key to the value network. Note that activities related to geoscientific work include the identification of mineral prospects and explicitly the link to the economic feasibility of extraction. Data management was identified as another key activity, as the processing of vast amounts of data is critical in all phases of exploration. To ensure conformity of project goals with public expectations, public relations and expectation management are critical. For the sustainable test site value network, the participants identified technology improvement related aspects, such as benchmarking of technologies, assessment of performance, comparison of technologies with benchmarks, and systems as critical. Beyond the technological perspective, experts identified three central sustainability efforts to be performed. First, the experts added monitoring of the social and physical environment at and in the surrounding area of the test sites as a key activity. Second, and for reasons of non-invasive innovation, the impact assessment on the social and physical environment is key. Third, and originating from the stakeholder workshop, participants argued for the integration of educational programs for the communities. Similarities between the sustainable test site business model and mineral exploration business model in public relations management were mentioned as complementary. Areas with significant differences between the sustainable test site and the common mineral exploration business model were found. These include the integration of the social and environmental perspective into the assessment of technologies, as well as monitoring the surrounding area. In addition, the sustainable test site explicitly directs its engagement activities towards communities and environmental groups.

**Key Resources:** The most important resources for exploration practice were identified as contract intangible assets (e.g., permits), precise knowledge and data about the subsurface, exploration technologies, as well as expertise in geosciences, engineering, land-use, and economics.

Both the sustainable test site and the traditional mineral exploration approach share many similarities. However, the sustainable test site considers knowledge about the social and physical environment as well as the social license to operate as a key resource.

**Key Partners:** Relevant partners in the traditional scenario originate from the resources needed and the activities performed. The most important partners are governments, expert associations, local and legal peers. Compared to the traditional scenarios, mining companies are included in the partner network. From the perspective of the experts, sustainability improvement requires the integration of NGOs, social and environmental advocates as well as independent agencies.

**Cost Structure:** In the traditional model, the main blocks listed are contract handling, cost of work and expert consultation. When it comes to the financial perspective of the sustainable value network, similarities between the sustainable test site business model and mineral exploration business model exist concerning the contracting expenses, cost of work and expert consultation. As drilling is not in the scope of the test sites, this aspect is not considered here. The experts further argued that additional activities and resources needed for social and environmental safeguards are reflected in the costs. Being more precise, the changes in the key activities, key resources, and increased efforts for managing the complexity of the partner network raise costs for highly skilled experts.

**Customer Relationship:** Customers of the traditional business model demand a transparent and trustful relationship on which to build their investment decision. The experts claim that the identified customer segments further value constructive journalism. In terms of the sustainable test site, customers also demand a transparent and trustful relationship. However, the reason is different. Since sharing sensible data between test site users and independent third parties for verification of a technology's performance is key to the test site approach, customer trust and transparency of operations are crucial factors to ensure a successful and long-term relationship. To ensure accountability, reliable protocols have to be provided, and unbiased information sharing, and processing have to be ensured. For customer relationships, the comparability of the results is limited.

**Customer Segments:** Mineral exploration targets mining companies and shareholders as their customers. In general, test sites intend to reach anyone interested in improving or assessing their

mineral exploration technologies, especially technology developers and mineral exploration companies of different sizes. Regarding the typical customer segment of sustainable business models, the experts argued that customers of the sustainable test site are more sensitive to present and future challenges of mineral exploration. Customers of the sustainable approach acknowledge the need to develop less invasive technologies and are willing to invest in activities to meet higher standards in terms of environmental impact and social acceptability.

**Distribution Channels:** Distribution channels in mineral exploration are online and offline media, conferences, stock exchanges, and case studies to communicate the technological performance of a technology. Similar to the traditional approach, sustainable test site customers are addressed via online and offline media, conferences, and case studies. Regarding community engagement, experts added direct communication by talks and calls to the list of channels.

**Revenue Streams:** As the revenue model strongly varies depending on the ownership structure, no decisive revenue model could be identified for either approach. The revenue streams listed in the table can be characterized as a summary of options. As a result, no conclusive statement about the revenue scheme can be made at this stage.

**Location-specific Adaptations:** Beyond the elements described above, three regional particularities came out of the results. Table 2 illustrates the location-specific adaptations identified by the workshop experts.

**Table 2.** Location-specific Adaptations.

| Country | Factor | Location-Specific Adaptations |
|---------|--------|-------------------------------|
| Finland | Key Activities<br>Key Partners | Special focus on environmental monitoring and protection<br>Locals (environmental and social stakeholders) |
| Germany | Key Activities<br>Value Proposition | Environmental protection<br>Promotion of non-invasive technologies |
| Spain | Key Activities | Education<br>Economic development<br>Continuous feedback to communities |

In Finland, environmental protection and its monitoring were of particular importance. A second aspect is the cooperation with the local Sámi community and reindeer herders, which is unique to the region and cannot be neglected. In Spain, economic development along with employment and education were raised as a focus. In particular, the test sites were seen as a way to deliver additional benefits by providing technical education opportunities. Germany was especially interested in the promotion of non-invasive technologies as the Erzgebirge region aims to decouple its reputation from potential negatives of its mining history.

## 6. Discussion

Using a business model approach, this paper illustrates a mechanism for the integration of sustainability efforts into technology development at test sites. By linking the test site concept with current industry practices, a perspective for practical transformation mechanisms for delivering industrial sustainability emerges.

The first RQ asked: How can sustainability measures be implemented in the design of test sites? Results of the study show that key activities, key resources, and key partners can drive sustainability at test sites. Concerning the named drivers, two main fields of improvement were identified. First, mechanisms to increase the sustainability of the technology itself. Second, factors to increase the sustainability of the test site project as a whole. The first scenario confirms the findings of [36–40] who argue that field tests can lead to performance improvements and increased visibility of a technology. The results extend the previous literature by adding the sustainability perspective. To this end, the paper suggests that by allowing only non-invasive technologies at the test sites and providing a

third-party evaluation, sustainable technologies can be promoted. In line with [40], the study shows that by enabling third-party evaluation and demonstration of non-invasive practices, companies can build a more sustainable image and lower market entry barriers. From a systems perspective, the study shows that sustainability at test sites can be promoted through the integration of non-traditional partners such as communities, NGOs and social and environmental advocates. The study adds to the existing literature by illustrating that the integration of social and environmental experts can improve the impact reduction potential and that higher social acceptance can be realized.

The second RQ asked: What insights can sustainable test sites provide for companies aiming to improve their sustainability efforts? The research process identified sustainability drivers, including: social and environmental expertise, monitoring of social and environmental conditions, social engagement, the integration of non-traditional partners such as NGOs, and environmental and social advocates [55,61]. While the latter is well established in the literature, the recognition that the inclusion of social and environmental experts can reduce the impact on the social and physical environment is new. The changes in key activities, key resources, and key partners increase costs through heightened coordination efforts. Additional resources such as social and environmental experts and their respective activities increase costs. Finally, to remain relevant despite higher costs, the targeted customer segments have to value sustainability requirements. While the sustainable narrative is increasingly pursued in mining [27,31], customers for mineral exploration activities are not yet fully sensitized. Figure 2 summarizes these insights.

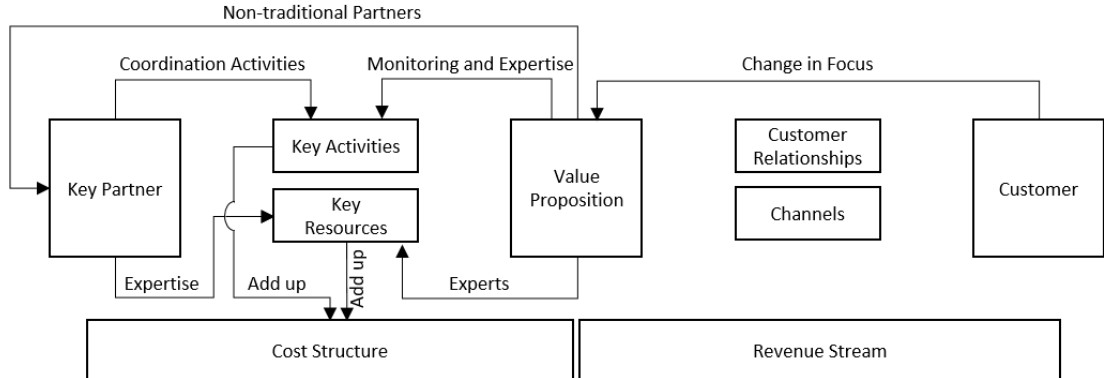

**Figure 2.** Sustainable Value Network.

Another important finding was that regional characteristics could influence the importance of identified factors.

In Finland, a strong focus on environmental protection and tight cooperation with locals was identified. Concerning the Spanish case, integration of knowledge gained, and the continuous communication with communities' matters. The emphasis here was on socio-economic development in the face of high local unemployment rates. In Germany, however, the focus was put on environmental protection and the promotion of technological development for non-invasive mineral exploration technologies.

Each regional particularity can be attributed to one of the business model factors identified in Table 1. However, the region-specific aspects indicate a need for aligning the weight of attributes with the regional context. This is where asset management comes into play, as the recognition of different elements must be complemented by a value-oriented approach that respects regional differences. These findings are in line [65], and expand the findings by arguing that resource prioritized asset management can increase sustainability.

This study set out to analyze the sustainable design of reference sites and the insights sustainable reference sites provide for companies aiming to improve their sustainability efforts. However, the results gained are subject to limitations. First, not all building blocks allow for drawing conclusions. Even though significant sustainability transformation potential was identified, customer, customer

relationships, channels, and revenue streams do not allow drawing conclusions from sustainable test sites to the mineral exploration business model. The main issue is that two different customer groups are considered in the sustainable test site approach and the general mineral exploration value network. Further research should be undertaken to investigate how sustainable test sites create systemic push effects for customers, customer relationships, channels, and revenue streams in the mineral exploration industry. A further limitation that emerged ex-post from the evaluation of results is the role of management systems.

The focus of the study is clearly on the structural environment, but the results reveal that concepts such as asset management were thus far only a side aspect. For future works, the management of the different building blocks, their control, and coordination could improve the value created throughout the chain of activities. Connecting the structural elements with asset management carries the potential to strengthen the impact on sustainability measures, which is not thoroughly addressed here because it only surfaced in the course of the evaluation.

Secondly, the sustainable test site approach is, to the authors' knowledge, the first of its kind. Results are, therefore, mostly conceptual and need validation once operations have picked up in earnest. In addition, even though the workshops provided first insights, they were held at an early stage of the INFACT project. Therefore, adjustments to the presented test site business model are likely. Finally, the technology development improvement potential was only identified qualitatively and was not validated through performance data analysis. This can be subject to future research and is indeed an objective in the INFACT project at present.

## 7. Conclusions

In conclusion, the implementation of social and environmental expertise has the potential to embed sustainability into technology development, increase the ambition of non-invasive practices, accelerate the introduction of sustainable technologies. In the environment, it also shows promise in reducing social and environmental risks through monitoring and consulting from social and environmental sciences. The study extends the literature by stating that sustainable technology development can benefit from the integration of social and environmental expertise and the respective stakeholders. Existing test sites can use the insights for shaping their own transformation, which can provide assistance in changing the sector as a whole from the bottom up.

**Author Contributions:** Conceptualization, M.K. (Michaela Kesselring) and F.W.; methodology, M.K. (Michaela Kesselring); validation, M.K. (Michaela Kesselring) and F.W.; formal analysis, M.K. (Moritz Kirsch), L.A., and R.G.; investigation, M.K. (Michaela Kesselring) and F.W.; resources, F.W.; data curation, M.K. (Michaela Kesselring) and F.W.; writing—original draft preparation, M.K. (Michaela Kesselring); writing—review and editing, M.K. (Michaela Kesselring), L.A., and R.G.; visualization, M.K. (Michaela Kesselring); supervision, F.W.; project administration, M.K. (Michaela Kesselring); funding acquisition, R.G. and L.A. All authors have read and agreed to the published version of the manuscript.

**Funding:** This research has received funding from the European Union's Horizon 2020 research and innovation program under grant agreement No 776487.

**Acknowledgments:** The authors gratefully acknowledge the contribution of partners on the INFACT project grant agreement No 776487.

**Conflicts of Interest:** The authors declare no conflict of interest. The funders had no role in the design of the study; in the collection, analyses, or interpretation of data; in the writing of the manuscript, or in the decision to publish the results.

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
