# Peer review of "Development of Sustainable Test Sites for Mineral Exploration and Knowledge Spillover for Industry"

_sustainability, doi:10.3390/su12052016_

Round 1
Reviewer 1 Report
The objective of the paper, understanding the relationship between sustainability and mineral exploration business practices, is nicely introduced in the paper. As well, the research questions developed fit the aim and allow for an interesting theoretical discussion on assessing sustainability at exploration test sites. But one of the biggest issues with the paper is the transition from theory to case/methodology and results.
The paper introduces INFACT as the case, which includes multiple test sites across Europe. The descriptions of the sites are relatively short and focused mainly on deposit type(s). I think some details on the different socio-political and environmental factors at each site would strengthen the case section. However, this might not be necessary since all references to these different sites end here. It appears that the input from both the experts within the project is an amalgamation of the findings from these test sites, but this generalization of the input from the experts (and from the participants from outside the project) does not allow the reader to discern what information has been gleaned from each test site or (worse) if it comes from previous or general knowledge of the experts. Some connections to the test sites should be included in the results section either in the table, as some type of summary of each site, or quotes or anecdotes from the participants. I believe this level of detail is needed to make the findings comparable to previous and future research.
Author Response
We thank the reviewer for the very constructive comments. With the provided help we were able to improve the manuscript. We implemented all the remarks accordingly.
The objective of the paper, understanding the relationship between sustainability and mineral exploration business practices, is nicely introduced in the paper. As well, the research questions developed fit the aim and allow for an interesting theoretical discussion on assessing sustainability at exploration test sites.
Thank you for your praise.
But one of the biggest issues with the paper is the transition from theory to case/methodology and results.
We agree with the comment. A transition paragraph, from theory to case/methodology has now been added to the theory section [L243-254].
The paper introduces INFACT as the case, which includes multiple test sites across Europe.
The descriptions of the sites are relatively short and focused mainly on deposit type(s). I think some details on the different socio-political and environmental factors at each site would strengthen the case section.
We agree with the reviewer, that socio-political and environmental factors of each site strengthen the case section. Information on socio-political and environmental factors at each site has been added to the methodology section [L264-269].
However, this might not be necessary since all references to these different sites end here.
References to each site have been integrated into the results [L426-433] and the discussion section [L470-483].
It appears that the input from both the experts within the project is an amalgamation of the findings from these test sites, but this generalization of the input from the experts (and from the participants from outside the project) does not allow the reader to discern what information has been gleaned from each test site or (worse) if it comes from previous or general knowledge of the experts.
We agree, that Section 3, Methodology needs clarification. For increased clarity, information related to the sample, their relationship to the project as well as their previous expertise has been added to the methodology section [L295-305].
While the ultimate goal of the workshops was the development of a general test site approach as a conglomerate of three test sites, we understand that a distinction between location allows for a deeper understanding of the topic and yields more precise results. We clarified the approach in Section 3, Methodology [L315-218] and included the identified distinctions between the test sites in the results and discussion section [L427-436].
Some connections to the test sites should be included in the results section either in the table, as some type of summary of each site, or quotes or anecdotes from the participants.
A table introducing the results at each individual test site has been added to the results section [L427-436].
I believe this level of detail is needed to make the findings comparable to previous and future research.
We agree with the reviewer and made the corresponding changes.
Reviewer 2 Report
Summary
This paper weighs in on possibilities for sustainable development in the mineral exploration industry. In doing so, it draws on results from the INFACT project. I thought the paper was insightful and will attract citations but I think a bit more clarification on what sustainability means in the context of mining and mineral exploration specifically is needed. I will provide some suggestions below. I do not think these revisions should take too long.
Recommendations
- The paper needs a thorough edit for grammar. The sentences are mostly complex and the paragraphs, by extension, convoluted.
- I think that there needs to be a well-referenced paragraph which defines sustainability (and maybe even CSR) in the mining or mineral extraction context. There has been a lot of discussion presented on this subject over the years. My thinking is that, even in the introduction, a small section could be inserted which clarifies to readers of Sustainability what this means (it is especially important, given the publication). Suggested references in need of reviewing:
- Barrera, E.B. Extractive industries and investor-state arbitration: Enforcing home standards abroad (2019) Sustainability (Switzerland), 11 (24), art. no. 6963, .
- Littleboy, A., Keenan, J., Ordens, C.M., Shaw, A., Tang, R.H., Verrier, B., Vivoda, V., Yahyaei, M., Hodge, R.A. A sustainable future for mining by 2030? Insights from an expert focus group (2019) Extractive Industries and Society, 6 (4), pp. 1086-1090.
- Fraser, J. Creating shared value as a business strategy for mining to advance the United Nations Sustainable Development Goals (2019) Extractive Industries and Society, 6 (3), pp. 788-791.
- Vivoda, V., Kemp, D. How do national mining industry associations compare on sustainable development? (2019) Extractive Industries and Society, 6 (1), pp. 22-28.
- Bainton, N.A., Owen, J.R., Kemp, D. Mining, mobility and sustainable development: An introduction (2018) Sustainable Development, 26 (5), pp. 437-440.
- Kemp, D., Owen, J.R. The industrial ethic, corporate refusal and the demise of the social function in mining (2018) Sustainable Development, 26 (5), pp. 491-500.
- de Lange, W., de Wet, B., Haywood, L., Stafford, W., Musvoto, C., Watson, I. Mining at the crossroads: Sectoral diversification to safeguard sustainable mining? (2018) Extractive Industries and Society, 5 (3), pp. 269-273.
- Hilson, A., Hilson, G., Dauda, S. Corporate Social Responsibility at African mines: Linking the past to the present (2019) Journal of Environmental Management, 241, pp. 340-352.
- Hilson, G., Murck, B. Sustainable development in the mining industry: Clarifying the corporate perspective (2000) Resources Policy, 26 (4), pp. 227-238.
3. Maybe some articulation of how mining is a lifecycle, and as this paper focuses on the exploration segment of mining, that these cases apply to it and it only. Again, I do not feel much is needed here, apart from a paragraph or two in the introduction, with references, which communicates to the reader very clearly that mining has distinctive phases.
I look forward to seeing the final version!
Author Response
The authors are thankful to the reviewer for the constructive critics. The resulting amendments improved the manuscript tremendously.
This paper weighs in on possibilities for sustainable development in the mineral exploration industry. In doing so, it draws on results from the INFACT project. I thought the paper was insightful and will attract citations but I think a bit more clarification on what sustainability means in the context of mining and mineral exploration specifically is needed.
Thank you very much for your commendation.
I will provide some suggestions below. I do not think these revisions should take too long.
Recommendations
- The paper needs a thorough edit for grammar. The sentences are mostly complex and the paragraphs, by extension, convoluted.
We took your recommendation seriously. English editing has been performed by a native speaker.
- I think that there needs to be a well-referenced paragraph which defines sustainability (and maybe even CSR) in the mining or mineral extraction context. There has been a lot of discussion presented on this subject over the years. My thinking is that, even in the introduction, a small section could be inserted which clarifies to readers of Sustainability what this means (it is especially important, given the publication).
We agree we the reviewer, that sustainability needs further clarification. In section 2 "Sustainability in mineral exploration,” a paragraph was added introducing the concept of sustainability and its meaning in the minerals sector. Current research gaps in this field were highlighted. [L102-127]. Company level considerations towards sustainability in the mining sector have been added, referencing Hilson et al. (2019) and Hilson et al. (2010) [L112-118].
Suggested references in need of reviewing:
- Barrera, E.B. Extractive industries and investor-state arbitration: Enforcing home standards abroad (2019) Sustainability (Switzerland), 11 (24), art. no. 6963, .
- Littleboy, A., Keenan, J., Ordens, C.M., Shaw, A., Tang, R.H., Verrier, B., Vivoda, V., Yahyaei, M., Hodge, R.A. A sustainable future for mining by 2030? Insights from an expert focus group (2019) Extractive Industries and Society, 6 (4), pp. 1086-1090.
- Fraser, J. Creating shared value as a business strategy for mining to advance the United Nations Sustainable Development Goals (2019) Extractive Industries and Society, 6 (3), pp. 788-791.
- Vivoda, V., Kemp, D. How do national mining industry associations compare on sustainable development? (2019) Extractive Industries and Society, 6 (1), pp. 22-28.
- Bainton, N.A., Owen, J.R., Kemp, D. Mining, mobility and sustainable development: An introduction (2018) Sustainable Development, 26 (5), pp. 437-440.
- Kemp, D., Owen, J.R. The industrial ethic, corporate refusal and the demise of the social function in mining (2018) Sustainable Development, 26 (5), pp. 491-500.
- de Lange, W., de Wet, B., Haywood, L., Stafford, W., Musvoto, C., Watson, I. Mining at the crossroads: Sectoral diversification to safeguard sustainable mining? (2018) Extractive Industries and Society, 5 (3), pp. 269-273.
- Hilson, A., Hilson, G., Dauda, S. Corporate Social Responsibility at African mines: Linking the past to the present (2019) Journal of Environmental Management, 241, pp. 340-352.
- Hilson, G., Murck, B. Sustainable development in the mining industry: Clarifying the corporate perspective (2000) Resources Policy, 26 (4), pp. 227-238.
We thank the reviewer for the references. The references were reviewed and added [L102-127].
- Maybe some articulation of how mining is a lifecycle, and as this paper focuses on the exploration segment of mining, that these cases apply to it and it only. Again, I do not feel much is needed here, apart from a paragraph or two in the introduction, with references, which communicates to the reader very clearly that mining has distinctive phases.
The mining lifecycle has been introduced. A clarification of the scope of the paper has been added into Section 2 Sustainability in Mineral Exploration [L119-127].
I look forward to seeing the final version!
Reviewer 3 Report
The paper is well prepared and tackles a topic of interest. However, the authors should discuss the impact of the contemporary complex operational and business environment which has a significant impact on ways how modern companies (mining and other) function. It the ultimate goal is to create real world conditions in test sites this aspect shall be taken into consideration. It is highly recommended that, in this regard, the authors consult some of suggested references below.
Moreover, a sustainability goal has to be considered in a broader enterprise context given that it is competing with numerous other factors and aspects. In this regard, the authors should discuss the role of a global asset management
Good Asset Management is a key enabler for organizations seeking to contribute to the achievement of the United Nations' Sustainable Development Goals (SDG). As stated in relevant literature in this area, Asset Management provides clarity of purpose and the management system to ensure that good intentions are turned into practical reality. Reliability only is unable to do so.
Effective and efficient organizations use a structured approach to their Asset Management in order to resolve competing priorities and ensure that long-term benefits are not sacrificed for more immediate needs. An Asset Management System (AMS) based on the ISO 55000 family of standards helps an organization to establish a coherent approach and coordinated delivery of appropriate resources and activities. It also incorporates monitoring and continual improvement elements to assure sustained achievement of the strategic objectives.
An effective and efficient AMS aligns an organization’s Corporate Social Responsibility (CSR) goals with daily activities and processes. Asset Management incorporates such goals into technical and financial decisions, to derive clear plans and activities. The AMS ensures performance delivery, monitoring and continual improvement to achieve the CSR goals at all levels.
Assets exist to provide value to the organization and its internal and external stake-holders. Asset Management focuses on the value that assets provide. Incorporating CSR goals that acknowledge the SDG into your value framework ensures that the Asset Management System incorporates the latter as success criteria. Asset Management realizes value by optimizing combinations of financial, environmental, social impact, risk management, quality of service and performance criteria throughout an asset’s life.
Good Asset Management gives assurance that assets will fulfill an organization’s CSR because it requires the following:
- Developing and implementing processes that connect the required purposes and performance of the assets to organizational objectives;
- Implementing processes for assurance of capability across all life cycle stages;
- Implementing processes for monitoring and continual improvement; and
- Providing the necessary resources and competent personnel to succeed.
It is strongly suggested that the authors discuss this aspect.
Suggested references
- Helbing, D. 2013. Globally networked risks and how to respond. Nature. 497: 51-59
- Katina, P. F., Pinto, C. A., Bradley, J. M., & Hester, P. T. 2014. Interdependency-induced risk with applications to healthcare. International Journal of Critical Infrastructure Protection, 7(1), 12–26
- Komljenovic, D., Abdul-Nour, G. and Popovic, N. (2015), An approach for strategic planning and asset management in the mining industry in the context of business and operational complexity, J. Mining and Mineral Engineering, Vol. 6, No. 4, pp.338–360
- Renn, O., Lucas,C., Haas, A., Jaeger, J. 2017. Things are different today: the challenge of global systemic risks, Journal of Risk Research; 1-16
- Stirling, A. 2010. Keep it complex, Nature, 468: 1029-1031
- Zio, E. 2016. Challenges in the vulnerability and risk analysis of critical infrastructures. Reliability Engineering and System Safety 152, 137–150
Author Response
We thank the reviewer for the positive remarks and the constructive review. We implemented all the aspects raised by the reviewer.
The paper is well prepared and tackles a topic of interest. However, the authors should discuss the impact of the contemporary complex operational and business environment, which has a significant impact on the ways how modern companies (mining and other) function.
We agree with the reviewer that introducing the dynamics of the mineral exploration sector is useful. Explanation of the business environment of mining, mineral exploration, and cross-sectoral functions has been added to Section 2 Sustainability in mineral exploration [L107-118].
It the ultimate goal is to create real world conditions in test sites this aspect shall be taken into consideration. It is highly recommended that, in this regard, the authors consult some of suggested references below.
We thank the reviewer for the references. The references were reviewed and added.
Moreover, a sustainability goal has to be considered in a broader enterprise context given that it is competing with numerous other factors and aspects.
A reference to the SDGs has been added to Section 2 Sustainability in mineral exploration [L107-121]
In this regard, the authors should discuss the role of a global asset management
The role of asset management has been added to Section 2 Sustainability in mineral exploration [L225-242]. The value of asset management in global operations was applied to the test site approach and discussed in Section 5 Discussion [L472-483].
Good Asset Management is a key enabler for organizations seeking to contribute to the achievement of the United Nations' Sustainable Development Goals (SDG). As stated in relevant literature in this area, Asset Management provides clarity of purpose and the management system to ensure that good intentions are turned into practical reality. Reliability only is unable to do so.
We agree that disregarding interdependencies between the connections between the structural elements of a business model is an oversimplification and does not do justice to the complexity of the mineral exploration business environment. A respective paragraph was included to discuss the limitations of a merely structural business creation approach (see Section 2.2 Sustainable Business Networks) [L224-233].
Effective and efficient organizations use a structured approach to their Asset Management in order to resolve competing priorities and ensure that long-term benefits are not sacrificed for more immediate needs. An Asset Management System (AMS) based on the ISO 55000 family of standards helps an organization to establish a coherent approach and coordinated delivery of appropriate resources and activities. It also incorporates monitoring and continual improvement elements to assure sustained achievement of the strategic objectives.
Asset management was defined base on the definition of the ISO 55000 family. The topic of prioritization of assets was included in Section 2.2 Sustainable Business Networks. We understand that in the case of three different test site locations, asset management is crucial to coordinate and the key resources and activities of a business model [L225-242]. Therefore, we added the topic of prioritization of business model building blocks through asset management into Section 5 Discussion [L479-485].
An effective and efficient AMS aligns an organization’s Corporate Social Responsibility (CSR) goals with daily activities and processes. Asset Management incorporates such goals into technical and financial decisions, to derive clear plans and activities. The AMS ensures performance delivery, monitoring and continual improvement to achieve the CSR goals at all levels.
We agree that asset management is critical to realize sustainability and strategic business goals. A corresponding section has been added to Section 2.2 Sustainable Business Networks [L234-242].
Assets exist to provide value to the organization and its internal and external stakeholders. Asset Management focuses on the value that assets provide.
We agree that in the case of three different test site locations, external, and internal operations have to be coordinated. A corresponding section has been added to Section 5 Discussion [L479-483].
Incorporating CSR goals that acknowledge the SDG into your value framework ensures that the Asset Management System incorporates the latter as success criteria. Asset Management realizes value by optimizing combinations of financial, environmental, social impact, risk management, quality of service and performance criteria throughout an asset’s life.
We agree with the reviewer, that asset management is a success criterion for corporate social responsibility. A corresponding section, of how asset management can promote sustainability at test sites has been added to Section 5 Discussion [L472-483].
Good Asset Management gives assurance that assets will fulfill an organization’s CSR because it requires the following:
- Developing and implementing processes that connect the required purposes and performance of the assets to organizational objectives;
- Implementing processes for assurance of capability across all life cycle stages;
- Implementing processes for monitoring and continual improvement; and
- Providing the necessary resources and competent personnel to succeed.
It is strongly suggested that the authors discuss this aspect.
We thank the reviewer for the detailed explanation and added the narrative described here into Section 2.2 Sustainable Business Networks [L227-244]. We further added a paragraph in Section 5 [L495-500], that recognizes that future research should emphasize asset management for mineral exploration.
Suggested references:
- Helbing, D. 2013. Globally networked risks and how to respond. Nature. 497: 51-59
- Katina, P. F., Pinto, C. A., Bradley, J. M., & Hester, P. T. 2014. Interdependency-induced risk with applications to healthcare. International Journal of Critical Infrastructure Protection, 7(1), 12–26
- Komljenovic, D., Abdul-Nour, G. and Popovic, N. (2015), An approach for strategic planning and asset management in the mining industry in the context of business and operational complexity, Mining and Mineral Engineering, Vol. 6, No. 4, pp.338–360
- Renn, O., Lucas,C., Haas, A., Jaeger, J. 2017. Things are different today: the challenge of global systemic risks, Journal of Risk Research; 1-16
- Stirling, A. 2010. Keep it complex, Nature, 468: 1029-1031
- Zio, E. 2016. Challenges in the vulnerability and risk analysis of critical infrastructures. Reliability Engineering and System Safety 152, 137–150
We thank the reviewer for the references. The references were reviewed and cited in Section 2.2 Sustainable Business Networks [L225-242].
Round 2
Reviewer 1 Report
Thanks for taking into account the comments raised in my review. I believe the most of the key issues have been addressed and the paper is ready for publication. In particular, the issue of differentiation between the sites has been addressed in a positive that references the theoretical model and makes the connections between methods, findings, and conclusions flow much better.
Reviewer 3 Report
The authors significantly improved the revised paper. It is recommended to accept it in the revised form.